# Deflamin Attenuated Lung Tissue Damage in an Ozone-Induced COPD Murine Model by Regulating MMP-9 Catalytic Activity

**DOI:** 10.3390/ijms25105063

**Published:** 2024-05-07

**Authors:** Elia Ana Baltazar-García, Belinda Vargas-Guerrero, Ana Lima, Ricardo Boavida Ferreira, María Luisa Mendoza-Magaña, Mario Alberto Ramírez-Herrera, Tonatiuh Abimael Baltazar-Díaz, José Alfredo Domínguez-Rosales, Adriana María Salazar-Montes, Carmen Magdalena Gurrola-Díaz

**Affiliations:** 1Instituto de Investigación en Enfermedades Crónico-Degenerativas, Instituto Transdisciplinar de Investigación e Innovación en Salud, Departamento de Biología Molecular y Genómica, Centro Universitario de Ciencias de la Salud, Universidad de Guadalajara, Sierra Mojada 950, Puerta peatonal 7, Col. Independencia, Guadalajara 44350, Jalisco, Mexico; elia.baltazar3265@alumnos.udg.mx (E.A.B.-G.); belinda.vargas@academicos.udg.mx (B.V.-G.); tbaltazar@hcg.gob.mx (T.A.B.-D.); jose.drosales@academicos.udg.mx (J.A.D.-R.); adriana.smontes@academicos.udg.mx (A.M.S.-M.); 2CECAV—Centro de Ciência Animal e Veterinária, Faculty of Veterinary Medicine, Lusófona University, Campo Grande, 376, 1749-024 Lisbon, Portugal; ana.isabel.lima@ulusofona.pt; 3LEAF—Landscape Environment Agriculture and Food, Instituto Superior de Agronomia, Universidade de Lisboa, 1349-017 Lisbon, Portugal; rbferreira@isa.ulisboa.pt; 4Laboratorio de Neurofisiología, Departamento de Fisiología, Centro Universitario de Ciencias de la Salud, Universidad de Guadalajara, Sierra Mojada 950, Puerta peatonal 7, Col. Independencia, Guadalajara 44350, Jalisco, Mexico; luisa.mendoza@academicos.udg.mx (M.L.M.-M.); mario.rherrera@academicos.udg.mx (M.A.R.-H.)

**Keywords:** pulmonary disease, lupin proteins, emphysema, peribronchial fibrosis, matrix metalloproteinase

## Abstract

Chronic obstructive pulmonary disease (COPD) is comprised of histopathological alterations such as pulmonary emphysema and peribronchial fibrosis. Matrix metalloproteinase 9 (MMP-9) is one of the key enzymes involved in both types of tissue remodeling during the development of lung damage. In recent studies, it was demonstrated that deflamin, a protein component extracted from *Lupinus albus*, markedly inhibits the catalytic activity of MMP-9 in experimental models of colon adenocarcinoma and ulcerative colitis. Therefore, in the present study, we investigated for the first time the biological effect of deflamin in a murine COPD model induced by chronic exposure to ozone. Ozone exposure was carried out in C57BL/6 mice twice a week for six weeks for 3 h each time, and the treated group was orally administered deflamin (20 mg/kg body weight) after each ozone exposure. The histological results showed that deflamin attenuated pulmonary emphysema and peribronchial fibrosis, as evidenced by H&E and Masson’s trichrome staining. Furthermore, deflamin administration significantly decreased MMP-9 activity, as assessed by fluorogenic substrate assay and gelatin zymography. Interestingly, bioinformatic analysis reveals a plausible interaction between deflamin and MMP-9. Collectively, our findings demonstrate the therapeutic potential of deflamin in a COPD murine model, and suggest that the attenuation of the development of lung tissue damage occurs by deflamin-regulated MMP-9 catalytic activity.

## 1. Introduction

Chronic obstructive pulmonary disease (COPD) is a leading cause of death worldwide, estimated to be responsible for approximately 3 million deaths every year, with a global prevalence of 10.3% [1] COPD is a heterogeneous lung condition characterized by chronic respiratory symptoms due to abnormalities of the airways (bronchitis) and/or alveoli (emphysema) that cause persistent, often progressive airflow obstruction [1]. The etiology of COPD is multifactorial, with ozone exposure (a component of air pollution) one of the major environmental risk factors for its development, especially in countries with lower sociodemographic index and in never-smoker COPD patients [2]. In murine models, acute and chronic ozone exposure triggers oxidative damage, inflammation, and lung tissue remodeling [3].

Pulmonary morphological changes underlie the main clinical manifestations of COPD and are represented by pulmonary emphysema and peribronchial fibrosis [1]. Pulmonary emphysema is related to inflammation [4,5] and cell senescence, which subsequently leads to pneumocyte death and degradation of the extracellular matrix (ECM) [6,7,8]. On the other hand, peribronchial fibrosis is mainly a response to the process of epithelial–mesenchymal transition (EMT) mediated by the disruption of the lung cell basement membrane, and it is widely regulated by the transforming growth factor β (TGF-β) signaling pathway [9,10]. Interestingly, regardless of the etiology and molecular mechanisms underlying these pathological changes, metalloproteinases, a family of zinc-dependent endoproteinases, have a predominant role in both processes: these molecules have been reported to be some of the major effector enzymes of COPD lung tissue remodeling [11]. In particular, MMP-9 or gelatinase B has been shown to be overexpressed and activated in both clinical [12,13,14,15] and experimental [16,17,18,19,20,21] COPD studies. MMP-9 is implicated in the breakdown of the ECM, which is the structural and respiratory dynamic support of the lungs [22]. Active MMP-9 hydrolyzes the ECM components, such as collagens, gelatin, and elastin, among other proteins [23]. Gelatinase B also plays a regulatory role in inflammation and fibrosis by catalyzing the activation of important molecules, such as pro-tumor necrosis factor-α (TNF-α) [24], pro-interleukin 1β (IL-1β) [25] and pro-TGF-β [26]. Additionally, MMP-9 has also been associated with mechanisms of COPD perpetuation [27], lung cancer progression [9,10], and acute exacerbations of COPD (AE-COPD), characterized by worsening of lung function and increased susceptibility to respiratory infections [28,29].

At present, COPD remains an incurable disease, and the objective of pharmacological treatment is the improvement of respiratory symptoms and the prevention of COPD exacerbations [1]. However, the restricted effectiveness of current therapies and the development of side effects of the drugs due to their prolonged use [22] justify research into new therapeutic options. Recently, in a study that investigated the MMP-9-inhibitory potential of different legumes, it was reported that a protein fraction isolated from white lupin (*Lupinus albus*) exerted the most potent capacity to inhibit MMP-9 [30]. This protein fraction was called deflamin, and it is obtained through a very specific extraction process, which includes, among other steps, subjecting the defatted lupin flour to boiling and acidifying the pH. Using different chromatographic, mass spectrometric, and electrophoretic procedures, a detailed characterization and analysis of deflamin by Mota et al., was published [31]. The authors concluded that deflamin is composed of a complex mixture of different polypeptides derived from the large subunit of δ-conglutin (or heavy chain), as well as a few small fragments of β-conglutin. The bioactive effect of deflamin has been investigated using in vitro models of colon adenocarcinoma [30,31] and in vivo models of ulcerative colitis [32] and tumorigenesis [33]. Among the main biological benefits derived from MMP-9 inhibition in these studies were a decrease in tissue damage and a limitation of markers of cancer progression, such as metastatic capacity and angiogenesis. Therefore, the present study aimed to evaluate the inhibitory effect of deflamin on MMP-9 in a COPD murine model induced by ozone exposure, as well as through an in vitro analysis, and subsequently predict by bioinformatic analysis a possible deflamin–MMP-9 interaction.

## 2. Results

### 2.1. In Vitro Inhibitory Effect of Deflamin on MMP-9 Activity Using the DQ-Gelatin Assay

Deflamin’s bioactive potential was tested in vitro before its use in the COPD in vivo model. Firstly, the isolated deflamin was verified by SDS-PAGE characterization as described previously [30]. Subsequently, the lyophilized extract was evaluated through the dye-quenched fluorescent gelatin assay (DQ-gelatin). For this experiment, several commercially available reagents were used, such as recombinant human MMP-9 (rMMP-9) and a substrate for this enzyme: a complex of gelatin bound to quenched fluorescein. Additionally, the polyphenol quercetin and 1,10-phenanthroline (an organic compound, chelator of metal ions with neuro- and nephrotoxic effects) were employed as metalloproteinase-inhibitory controls. The principle of this technique allowed us to evaluate the gelatinase-inhibitory capacity exerted by the different compounds analyzed. Active rMMP-9 is capable of cleaving the bond between gelatin and fluorescein, causing fluorescence emission, which is measured through a fluorescence reader. A higher degree of gelatinase activity was reflected by high values of fluorescence emission, while a higher degree of gelatinase inhibition was reflected by low values of fluorescence emission. In the DQ-gelatin assay, the values of the rMMP-9 inhibition exerted by deflamin, quercetin, or phenanthroline were calculated relative to rMMP-9 gelatinase activity without the addition of any inhibitor. Therefore, the values of the fluorescence obtained in the wells with rMMP-9 without inhibitor (positive control), were considered to represent 100% of rMMP-9’s gelatinase activity. Correspondingly, the activity or inhibition values of any of the substances analyzed were calculated as a percentage relative to this 100%. The deflamin concentration used in the DQ-gelatin assay was determined based on previous studies that evaluated the dose response of deflamin for this type of analysis. It was ensured that even at the selected dose, cell viability was not impaired [31,34]. Quantitative results showed that deflamin at 98 µM concentration significantly inhibited 60% of the rMMP-9 gelatinolytic activity relative to the enzyme control without the addition of any inhibitor (*p* < 0.0001, Figure 1). With regard to the inhibitors, quercetin at a concentration of 750 µM and phenanthroline at 0.5 µM inhibited up to ≈80% of the gelatinase activity of rMMP-9 (*p* < 0.0001).

### 2.2. Attenuation of Lung Damage after Deflamin Treatment in Chronic Ozone-Exposed Mice

Histological assessment was performed to evaluate the morphological changes in lung tissue after chronic ozone exposition and the effect of deflamin treatment (20 mg/kg BW) in chronic ozone-exposed mice. H&E staining was used to evaluate the development of lung emphysema by quantifying the mean linear intercept (Lm) (Figure 2). Additionally, employing Masson’s trichrome staining, peribronchial fibrosis was also evaluated (Figure 3).

Lung tissues from the air-exposed group (control) did not demonstrate histological damage, with preserved alveolar and bronchial structures without cellular signs of either inflammation or fibrosis. In contrast, in the group exposed to ozone without treatment, widening of the alveolar septa and micro-collapse of the lung tissue were observed, alternating with loss of alveolar epithelium, tissue necrosis, and the presence of cellular debris. In addition, vascular alterations such as congestion, hemorrhage, and in some cases formation of capillary thrombi were demonstrated. Regarding cell types, no presence of lymphocytes was observed, but macrophages and fibroblasts were identified, as well as type 2 pneumocyte hyperplasia. Masson’s trichrome staining allowed the identification of collagen deposition not only in the peribronchial area but also in the lung parenchyma, as well as the thickening of smooth muscle in the bronchial walls, with a consequent reduction in the bronchiolar lumen. Regarding the treated group, deflamin exerted a protective effect, because although lung tissue alterations were similar to those exposed only to ozone, they were found to be less severe and extensive, with most of the alveolar spaces preserved. Quantification of Lm and collagen deposition turned out to be similar in the air group and the deflamin-treated group, whose results were statistically lower compared to the group exposed only to ozone (*p* < 0.0001).

### 2.3. Restoration of MMP-9 Protein Expression in Lung Tissue from Treated Animals

To relate the results obtained in the DQ-gelatin assay to the evident improvement in histopathological damage in the treated group, the protein expression of MMP-9 was evaluated in the experimental groups by Western blotting. A significantly higher MMP-9 expression was observed in the group exposed only to ozone in comparison to the group exposed to air (1.7-fold increase, *p* < 0.05). On the other hand, although a statistically significant difference was not reached between the treated group and the group exposed only to ozone, mice administered deflamin exhibited a downward trend in the expression of gelatinase B (Figure 4).

### 2.4. Inhibition of the MMP-9 Activity in Deflamin-Treated Mice

To evaluate the MMP-9 gelatinolytic activity in the lung tissue samples, zymography assays were performed. In the zymogram gel, different active forms of MMP-9 and MMP-2 (Figure 5a) were resolved. Densitometry analysis showed that deflamin significantly decreased the enzymatic activity of MMP-9 compared to the ozone group (*p* < 0.05), showing similar activity to the air control group (Figure 5b).

### 2.5. Prediction of Deflamin Fragments and MMP-9 Interaction

Molecular docking analysis was performed to corroborate our experimental results, as well as to contribute to the understanding of the deflamin inhibition mechanism on MMP-9 catalytic activity. Following Lima et al. [30] and Mota et al. [34], deflamin extraction was established through sequential harsh treatments and a final precipitation step to obtain a water-soluble bioactive protein fraction, which exerted higher MMP-9 inhibition. After several characterization methods, it was determined that deflamin derives mostly from the δ-conglutin large subunit, as well as a few small fragments of β-conglutin [31]. Furthermore, in deflamin, no fragments were found that corresponded to the minor subunit of δ-conglutin. Likewise, it is important to mention that due to the complexity of the proteins present in legumes, the crystallized structure of δ-conglutin is not available. For this reason, we performed homology modeling using the primary sequence of the δ-conglutin large subunit. On the other hand, although the fragments from δ-conglutin and β-conglutin can behave as an oligomer, the mechanism of interaction between these subunits has not been described [31]. Therefore, the δ-conglutin large subunit of *L. albus* species was selected and docked with a crystallized MMP-9 structure. Molecular docking results showed a total of 51 protein–protein interactions. Among them, nine hydrogen bonds (H-bonds) were identified (Table 1). Six of them were located in the MMP-9 catalytic domain and the rest were among the fibronectin domains. Hydrogen bonds maintained a distance of around 3.0 Å and most of them were of the type O-H···O and N-H···O, which are two of the stronger H-bonds. Notably, ASN71 residue from the δ-conglutin large subunit was found to establish two H-bonds with the MMP-9 catalytic domain (TYR128 and ARG158 residues).

Additionally, the rest of the protein–protein interactions were screened, considering those residues maintained the highest number of different interactions. From this screening, six main interactions were retrieved (Table 2), which were characterized by maintaining a distance from 2.6 to 4.8 Å, and most of the residues involved were polar charged and polar uncharged, contributing to the electrostatic interactions, such as salt bridges and induced dipole–dipole, respectively. On the MMP-9 side, the most important residues were TYR160 and ARG134, with eleven and five interactions, respectively. TYR160 established the highest number of different interactions, including one H-bond. Therefore, ASN71 from the δ-conglutin large subunit and TYR160 from MMP-9 could be considered relevant residues to stabilize the molecular interactions between both proteins. Graphical interactions between the δ-conglutin large subunit and MMP-9 are presented in Figure 6.

## 3. Discussion

The present study demonstrated the therapeutic potential of deflamin to attenuate histopathological changes in the lung architecture of mice exposed to ozone through its ability to regulate the catalytic activity of MMP-9. To determine the dose of deflamin in the murine COPD model, we referenced a previous study that used deflamin in a mouse model of ulcerative colitis, administering a dose of 15 mg/kg body weight orally. However, since the expected effect in that study was on the gastrointestinal tract and our target organ is the lungs, we considered increasing this dose for the present study. In our murine COPD model, we found that chronic ozone exposure significantly enlarged alveolar spaces, producing pulmonary emphysema, and induced a significant increase in the percentage of peribronchial fibrosis. These results are in agreement with those previously published in several studies, which implemented comparable COPD model conditions [5,35,36,37]. Interestingly, the development of these lung tissue alterations induced by ozone exposure was attenuated by deflamin administration, with the resultant tissue after treatment almost resembling normal lung architecture. It is important to note that attenuation of these morphometric parameters is consistently related to a functional improvement in respiratory capacity, as has been previously reported by Li et al. [5] and Jiang et al. [38].

The observation of MMP-9 protein expression indicates the cells’ capacity to carry out protein synthesis, as well as the secretion of this enzyme into the extracellular milieu [39,40]. In our study, we found significantly higher MMP-9 protein expression in the lungs of mice exposed to ozone compared to those exposed only to air, representing one of the effects of the oxidative and proinflammatory roles of ozone in lung tissue [3,5]. These results are in line with data published by other groups, who used a COPD model after exposure to ozone [41] or to tobacco [42,43]. Even though no statistical difference was observed between the control group exposed to ozone and the control group treated with deflamin, a downward trend in MMP-9 protein expression was detected in the latter.

The relevance of higher MMP-9 protein expression only makes sense if this enzyme exhibits higher catalytic activity in the lung parenchyma. Therefore, we assessed MMP-9 gelatinase activity in the experimental mouse groups by zymographic assays. We found a significant increase in gelatinase activity in mice exposed to ozone compared to those exposed only to air. Similar data were found in other studies using COPD models due to exposure to either ozone [41] or electronic cigarettes [44]. However, deflamin administration significantly decreased the MMP-9 gelatinase activity. A valuable aspect of this observation is that deflamin did not eliminate gelatinase activity, but only modulated it until it reached a physiological level. This is favorable, as reported by Yoon et al. [41], who observed enhanced epithelial damage in animals exposed to ozone, as well as a greater expression of proinflammatory proteins in MMP-9 deficient mice compared to wild-type (WT) mice. This demonstrates the therapeutic convenience of not eliminating the gelatinase B activity, but only of reestablishing it at the physiological level.

These deflamin biological effects could be explained by a time-coordinated interaction between enteral deflamin absorption and MMP-9 cellular release. Regarding the gelatinase B production, it has been reported that MMP-9 increases in bronchoalveolar lavage (BAL) from the first hour of ozone exposure [36]. Additionally, Paemen et al. [45] identified an increase in MMP-9 in blood plasma 30 min after exposing a group of baboons to lipopolysaccharides (LPS). Furthermore, they demonstrated that gelatinase B release remains constant for up to 6 h after cessation of the inflammatory insult. We hypothesize that in our experimental groups, neutrophils could be the main source of MMP-9 in lung samples. Neutrophils are capable of secreting gelatinase B in the presence of proinflammatory stimuli within minutes to an hour. On the other hand, macrophages produce MMP-9 de novo synthesis, which involves its secretion up to 6 to 12 h later [39,46]. The above hypothesis is based on the fact that zymographic analysis allowed us to identify different MMP-9 structures present in lung samples. This is possible because denaturing (although not reducing) conditions are used during the first stage of the zymography assay [47]. This allow us to visualize the catalytic activity of different MMP-9 structures, such as homo-oligomers (>220 kDa), bound to lipocalin (NGAL, ≈125 kDa), in monomeric form (92 and 82 kDa), and the ≈60 kDa form, which is produced by MMP-3 activation. It is worth mentioning that all these structures are produced mainly by neutrophils, which traffic to the lung tissue through a chemotactic effect. There, they secrete MMP-9 free of tissue inhibitor of metalloproteinase (TIMP) 1, its main endogenous inhibitor, and through preformed enzyme vesicles [39,46].

Concerning the administration of the treatment, Wojtowicz et al. [48] found that peptides derived from an orally supplied protein can be released into the blood circulation within 1 to 2 h after administration, resulting in a half-life of up to 4 to 5 h. With respect to our experimental design, it is important to highlight that deflamin is resistant to acid pH [31], which is an advantage to diminish the effect of in vivo digestion. Additionally, evidence of alternative enteral absorption of bioactive peptides has been reported, especially through a pathway known as paracellular diffusion. Such transport involves the passage of small amounts of peptides through the tight junctions that connect the apical and basolateral membranes of enterocytes. Lunasin, a bioactive peptide of 43 amino acid residues and present in lupin and soybean, has been reported to be transported into the blood circulation through this mechanism [49]. However, further analyses are required to verify this possible transport mechanism in the case of deflamin. Taking together this information, it is hypothesized that deflamin could inhibit the released MMP-9 in lung tissue around 2 h after ozone exposure. This view is supported by the results of the MMP-9 protein expression assay, as well as by the data on the half-life of gelatinase B. Based on Western blotting, it is only possible to identify an immediate effect of the MMP-9 and deflamin interaction. On the other hand, the results of the histological and enzymatic evaluations suggest that the beneficial effect of deflamin could be cumulative, as each ozone exposure (12 exposures total) was followed by the administration of deflamin, thus potentially counter-regulating the activity of MMP-9 to a greater extent. Furthermore, this cumulative effect could be due to a decrease in the production of matricines (peptides resulting from the extracellular matrix -ECM- degradation) and their proinflammatory effect [27,50].

Before performing the experiments with the animal model, an in vitro assay with a gelatinase fluorescent substrate (DQ-gelatin) was carried out, in order to evaluate the inhibitory capacity of deflamin on a human active rMMP-9. Our results demonstrated that deflamin reduced gelatinase activity by 60%. It is important to note that the comparison of our results with previous studies is dependent on the variability of the conditions used for each DQ-gelatin assay, or in some cases, certain relevant data, such as the concentration of the reagents used, which were not specified. As a reference, Saragusti et al. [51] achieved 80% inhibition of MMP-9 gelatinase activity using a 300 µM quercetin concentration. However, it was not possible to determine the molar concentration of the enzyme they used, because they did not specify the final reaction volumes. As another reference to this finding, in the study published by Vandooren et al. [52], different compounds were evaluated using the DQ-gelatin assay. The authors identified the three most efficient inhibitors of MMP-9: a phenylalanine derivative (batimastat), an aromatic ether compound (SB-3CT), and the polyphenol epigallocatechin-3-gallate (EGCC). These molecules inhibited gelatinase activity by 94%, 91%, and 33%, respectively. It is worth noting that in that study, they used a sixfold-lower MMP-9 concentration than in with our assay conditions, and this definitively could impact the percentage of MMP-9 inhibition. Furthermore, of these compounds, only batimastat can be considered comparable to deflamin, due to its chemical nature as an amino acid derivative. On the other hand, it is important to highlight that the results of this in vitro analysis must always be corroborated through in vivo studies, since the expected biological effect is not always confirmed in animal models. Such is the case for SB-3CT, which when administered in an Alzheimer’s murine model (25 mg/kg) failed to reduce the in vivo MMP-9 activity or the elimination of beta-amyloid plaques [53]. Therefore, our in vitro and in vivo assays allowed us to confirm the following three observations: (1) the effect of deflamin obtained at the in vitro level agrees with the inhibition of gelatinase B activity in the COPD animal model; (2) this effect was translated into a biological benefit (attenuation of pulmonary emphysema and peribronchial fibrosis); and (3) deflamin is capable of interacting with and inhibiting both human and mouse MMP-9. This is useful, as it allows us to propose that deflamin could generate a similar beneficial effect in humans.

In addition, using in silico molecular docking analysis, we investigated the interaction between the main deflamin fraction (δ-conglutin large subunit) and MMP-9 using data from crystallized human gelatinase B [54]. Although these bioinformatic data are preliminary and will require updating, the results that we obtained may contribute to the elucidation of how deflamin and MMP-9 interact. It should be noted that the analysis method employed (HDOCK) is limited to determining interactions between two rigid molecular structures, i.e., we cannot determine if there were structural changes once these potential interactions had occurred. However, this is a common limitation of all state-of-the-art approaches employing this analysis.

Docking analysis suggested an interaction of deflamin, both with the catalytic domain and the fibronectin domain of MMP-9, through hydrogen bonds and electrostatic interactions. Although none of these interactions involved the residues that conform to the enzyme active site (Zn^2+^ ion tetrahedrally coordinated by His401, His405, His411, and Cys99 residues) [23], it may be that these interactions compromise the MMP-9 three-dimensional and functional structure and/or interfere with binding to enzymatic substrates. This finding could explain the partial inhibition of gelatinase B observed in our in vitro and in vivo models. Intriguingly, most bioinformatic analyses of the interaction of MMP-9 with inhibitors have been especially focused on the use of secondary metabolites, such as polyphenols [51,55,56]. On the other hand, the use of proteinaceous MMP-9 inhibitors represents several advantages, one of which is the fact that its interaction surface is broader compared to other small molecules, which require higher affinity and more specificity to interact with MMP-9. That could increase their selectivity and decrease the risk of adverse effects [57].

Considering previous research on the use of deflamin to inhibit MMP-9 in other animal models of disease [30,31,32,33], we investigated for the first time the biological effect of deflamin in a murine COPD model as a novel approach to explore its therapeutic potential in this disease. Our results demonstrate its efficacy with respect to this objective. However, further studies are necessary to evaluate if deflamin exerts an effect at the antioxidant or anti-inflammatory level. Finally, determining if deflamin is modified at the hepatic level or by the microbiota, as well as the absorption site, will require further investigation and the use of protein pharmacodynamic and pharmacokinetic models.

## 4. Materials and Methods

### 4.1. Plant Material and Isolation of Deflamin

Certified *L. albus* seeds were kindly provided by Dr. Edzard van Santen (College of Agriculture, Auburn University, Auburn, AL, USA). Dehulled lupin seeds were ground and subjected to the Soxhlet method to obtain defatted flour to proceed with deflamin extraction as previously reported [Error! Reference source not found.], with some modifications. Briefly, 20 g ± 0.1 g of lupin flour was dissolved in 50 mM Tris-HCl buffer, pH 7.5 (1:10, *w*/*v*) and maintained under constant stirring overnight at 4 °C. After this step, the suspension was centrifuged at 13,500× *g* for 30 min at 4 °C and the pellet discarded. Next, the supernatant was boiled for 10 min and then centrifuged at 13,500× *g* for 20 min at 4 °C. Subsequently, the supernatant was adjusted to pH 4.0 and then centrifuged at 13,500× *g* for 20 min at 4 °C. Afterwards, the pellet was resuspended in 25 mL of 40% (*v*/*v*) ethanol, and the suspension obtained was centrifuged at 13,500× *g* for 30 min at 4 °C. The resultant supernatant was mixed with pure ethanol until reaching a final alcohol concentration of 90% (using 5 mL of pure ethanol per each milliliter of 40% ethanol used in the previous step). This dissolution was left overnight at −20 °C. The following day, all the content was centrifuged at 13,500× *g* for 30 min at 4 °C and the pellet obtained was resuspended in the smallest possible volume of double-distilled water (DDW). The extract obtained, containing isolated deflamin, was stored in microtubes at −80 °C or lyophilized in a FreeZone 4.5 FreezeDry (Labconco, Kansas City, MO, USA) for 2 h, at −50 °C and 0.036 mbar.

### 4.2. Sodium Dodecyl Sulfate–Polyacrylamide Gel Electrophoresis

Deflamin isolation was verified through one-dimension electrophoresis following the protocol described by Laemmli [58], with some modifications as per Mota et al. [34]. In summary, we prepared a 17.5% polyacrylamide resolving gel and a 4% polyacrylamide stacking gel, and performed the electrophoresis in a Mini-Protean^®^ Tetra cell system (BioRad, Segrate, Milan, Italy) at 200 V. To improve the resolution of the protein bands in the gel, an anode buffer was added with 0.1 M sodium acetate. Samples (aqueous extract) were treated with 2x Laemmli sample buffer (Biorad, Hercules, CA, USA; 161-0737) under reducing conditions and heated at 90 °C for 3 min. The molecular weight (MW) of deflamin was compared with a protein ladder (Thermo Scientific PageRuler Prestained Protein Ladder, 26616, Darmstadt, Germany). Gels were fixed for 1 h in 50% (*v*/*v*) methanol and 2% (*v*/*v*) of ortho-phosphoric acid, then washed three times with DDW and stained overnight in 0.5% (*w*/*v*) Coomassie brilliant blue G-250 (BioRad, Segrate, Milan, Italy) in 34% (*v*/*v*) methanol, 17% (*w*/*v*) ammonium sulfate, and 2% (*v*/*v*) ortho-phosphoric acid. Discoloring was carried out with DDW until protein bands were clearly visible against a clear background.

### 4.3. Fluorogenic DQ™-Gelatin Assay

For this assay, a 96-well black microplate (Greiner Bio-one, 655076; Kremsmünster, Austria), a recombinant human MMP-9 (rMMP-9; Sigma, M8945, St. Louis, MO, USA), substrate dye-quenched fluorescent gelatin (DQ-gelatin, DQ-G), 1,10-phenanthroline from an Invitrogen kit (D12055; Eugene, OR, USA), and quercetin obtained from our laboratory (a kind gift from Dr. Adriana Salazar) were used. For activation, the enzyme was dissolved in a 1:1000 ratio in a buffer containing 100 mM Tris base, 100 mM NaCl, 100 mM CaCl_2_, and 0.05% Tween 20. Immediately, it was incubated at 37 °C for 5 h, before its use. As gelatinase-inhibitor control, 0.5 µM 1,10-phenanthroline (final concentration) was used and prepared according to the manufacturer’s instructions. Additionally, 750 µM quercetin was used as inhibitory control, prepared according to Saragusti et al. [51]. Subsequently, 0.6 nM MMP-9 was added per well (40 µL of the active enzyme for a final volume of 180 µL), to which 98 µM (3 mg/mL) deflamin diluted in DDW was added, and the plate was incubated at 37 °C for 15 min. Next, the DQ-G substrate, which was prepared before use, was loaded according to the manufacturer’s instructions and used at a final concentration of 5.6 µg/mL (40 µL of prepared reagent). Subsequently, the plate was incubated again for 30 min at 37 °C. Afterward, the plate was analyzed in a fluorescence reader (BioTek Synergy HTX Multimode Reader, Agilent; Santa Clara, CA, USA) at λexcitation = 495 nm and λemission = 520 nm, and then three times every hour. The results considered were those obtained 2 h after the first reading, as they showed the stability of the reaction. To perform a test, each sample and controls were analyzed in triplicate. In each experiment, additionally to the sample (deflamin), two negative controls (deflamin without enzyme—only activation buffer and DDW without enzyme) and positive control (DDW with enzyme) were included. All data were corrected by subtraction of their respective negative controls. Next, the gelatinase activity was calculated as a percentage of the control of the enzyme (enzyme without any inhibitor, equaled to 100% of the gelatinase activity).

### 4.4. Animals and Experimental Design

Pathogen-free, seven-week-old male C57BL/6 mice (18 g ± 2 g) were provided by the bioterium of the Universidad de Guadalajara. Animals were housed within filter-topped cages, fed a standard rodent chow diet and water ad libitum, and were maintained under constant conditions of temperature (24 ± 2 °C), humidity (50–60%), and 12 h dark/light cycles. Twenty-one mice were received one week before the established age and subjected to an adaptation period, which consisted in their transference and allocation in the ozone exposure chamber (without ozone supply) for 3 h. Afterwards, mice were returned to their respective cages. Mouse manipulation was always performed by the same person to avoid stress. Mice were randomly distributed into three experimental groups of 7 mice each: (1) air control, (2) ozone, and (3) ozone + deflamin. Ozone exposure was performed considering the technical specifications described by Nery-Flores et al. [59]. The ozone and ozone + deflamin groups were exposed to ozone generated with an ozonizer (Certizon C100; Sander Elektroapparatebau GmbH, Uetze, Hannover, Germany) that was connected to an oxygen concentrator. Ozone was transferred to the premix chamber and a constant flux of ozone-free air was also applied. The mixed flux was transferred into a hermetic acrylic chamber (65 × 25 × 45 cm L/H/D), and animals were exposed twice a week for 6 weeks for 3 h at a concentration of 2.5 parts per million (ppm). Ozone concentration was continuously monitored by an ozone detector (ES-600, Ozone Solutions Inc., Hull, IA, USA). On alternate days, control mice (air control) were exposed to normal air (ozone-free air, ozone concentration under 0.04 ppm) for 3 h twice a week for 6 weeks. The ozone + deflamin group was treated by oral gavage with deflamin (20 mg/kg BW) dissolved in 100 µL of injectable water 30 min after ozone exposition. The air control and ozone groups received the diluent in the same conditions.

### 4.5. Tissue Samples

At the end of the experimental period, mice were anesthetized with i.p. Zoletil^®^ 50 (120 mg/kg BW; Virbac, Carros, France). Next, a certified pathologist dissected the platysma and anterior tracheal muscles to remove the lungs, washed them in sterile saline, and dissected them considering the following. In a preliminary study for the standardization of the murine model of ozone-induced COPD, it was identified that there is no significant difference between the left and right lungs regarding the development of emphysematous damage and peribronchial fibrosis. Derived from this observation, for the present study, the histological evaluation was performed on a portion of the right lung, and protein extraction was carried out on a homogenate between the right and left lung.

### 4.6. Lung Histological Assessment

The right lung fragments of the mice were fixed in 4% paraformaldehyde and then embedded in paraffin. Subsequently, sections 4 μm thick were cut and stained with hematoxylin and eosin (H&E) or Masson’s trichrome staining to be analyzed by light microscopy (Primo Star 3, Carl Zeiss, Jena, Germany) coupled with an Axiocam 208 color camera and ZEN Microscopy Software, version 3.9 (Carl Zeiss, Jena, Germany). H&E-stained slides were captured at 10× magnification and used to evaluate the magnitude of the alveolar spaces (degree of emphysema) by quantifying the mean linear intercept (Lm). For this, free areas of blood capillaries and respiratory ducts were considered, and a grid of 10 × 10 lines was placed over each field. At least three fields were considered per microscope slide. The Lm was calculated by dividing the total horizontal length of all the lines by the total number of alveolar septa that intercepted the grid lines [60,61]. Masson’s trichrome staining was used to measure the peribronchial fibrosis in lung tissue under 10× magnification. Cross sections of small airways (<2 mm in diameter) were considered, and collagen deposition was calculated as the percentage of the area stained in blue in terms of the total area of the airway [62].

### 4.7. Western Blot Analysis

Lung tissue (~200 mg) was homogenized in 350 μL of T-PER lysis buffer (tissue protein extraction reagent, ThermoFisher, USA) added with a complete protease inhibitor, mini EDTA-free protease inhibitor cocktail (Roche, Darmstadt, Germany). The total proteins extracted were quantified in each sample using the bicinchoninic acid method (BCA, Pierce BCA Protein Assay, ThermoFisher Scientific, NY, USA). Lung protein samples (20 μg per well) were separated under denaturing and reducing conditions on 8–12% SDS-PAGE gels. Next, the proteins were transferred with a semidry blotter to a polyvinylidene fluoride membrane (Thermo Scientific, 88518; Rockford, IL USA). Subsequently, the membrane was blocked for 1 h with a 5% defatted milk solution in TBS-T buffer (1X Tris buffer saline-Tween 20). Afterward, the membranes were incubated for 12 h at 4 °C with the anti-MMP-9 monoclonal primary antibody (Ab228402, anti-mouse; Abcam, Cambridge, MA, USA), 1:1000 or anti-beta-actin (Ab115777, anti-mouse; Abcam), 1:1000, diluted in blocking solution. The next day, the membranes were exposed for 1 h at 4 °C to the secondary antibody linked with horseradish peroxidase (Ab97051, anti-rabbit; Abcam), 1:5000. Lastly, the detection of the bands was carried out with a UVP ChemStudio Analytik image documenter (Analytik JENA; CA, USA) using the ECL kit (Millipore, Billerica, MA, USA) to generate the chemiluminescence reaction. VisionWorks Software (version 8.21) was used to analyze the results by densitometry.

### 4.8. Gelatin Zymography

Protein samples (20 μg per well) were separated by SDS-PAGE on a separating gel made of 8% acrylamide (*w*/*v*) and 0.1% gelatin (Sigma-Aldrich, G1890; St. Louis, MO, USA). The samples were not subjected to heating or reducing agents. The samples were mixed with a 5x loading buffer composed of 0.313 M Tris-HCl, pH 6.8, 10% (*w*/*v*) SDS, 50% glycerol (*v*/*v*), and 0.05% (*w*/*v*) bromophenol blue. Electrophoresis was performed at 20 mA for 50 min, and at the end, the gel wase washed three times with 50 mL of 2.5% (*v*/*v* in DDW) Triton X-100 buffer for 15 min each time. The renaturing buffer was discarded and replaced with 30 mL of zymography buffer (pH 7.4, 50 mM Tris, 5 mM CaCl2) and maintained under gentle agitation at room temperature for 30 min. The developing buffer was then replaced with fresh 50 mL of zymography buffer and incubated at 37 °C for 18 h. Subsequently, the gel was stained with the Coomassie brilliant blue previously described for SDS-PAGE. The gel was destained with destaining solution until white bands against a blue background were clearly visible [63,64]. VisionWorks Software (version 8.21) was used to analyze the results by densitometry.

### 4.9. Docking of Deflamin Fragments and MMP-9

As a prior step to the docking analysis, the three-dimensional structure of the larger deflamin fragments was modeled by homology from full sequencing of the δ-conglutin large subunit of *L. albus* (Q333K7 UniProtKB code) using the I-TASSER program [65,66,67]. Standard parameters without steric restrictions were implemented. From the five models retrieved, the selected model based on C-score (0.03) was further refined using the GalaxyRefine server [68,69] and validated through MolProbity [70]. The X-ray crystallized structure of a human MMP-9 (2.5 Å resolution; containing the pro-domain, the catalytic domain, and three fibronectin type II domains, PDB ID: 1L6J) [54] was downloaded from the Protein Data Bank (https://www.rcsb.org/, accessed on 6 January 2024). The PDB formats of the deflamin fragments (δ-conglutin large subunit) and MMP-9 were docked using the HDOCK server [71]. The interaction models were evaluated according to the docking and confidence scores, resulting in −209.8 and 0.7678, respectively, for the selected model. Next, this model was used to identify and visualize hydrogen bonds using UCSF Chimera software (version 1.17.3) [72]. Subsequently, the rest of the interactions retrieved from the HDOCK server were screened in order to identify those amino acid residues that established the largest number of different interactions in order to detect the residues with greater relevance in the interaction of both proteins. The six most important interactions (of the non-hydrogen-bonding type), were described based on the nature of each pair of residues, as well as the interaction distance between them.

### 4.10. Statistical Analysis

Data are expressed as means ± standard error of the mean (SEM). Statistical significance between groups was determined by one-way ANOVA and Tukey’s post hoc test. GraphPad Prism 6 version 6.01 (GraphPad Software, Inc.; CA, USA) was used for statistical analysis, and *p* < 0.05 was considered significant.

## 5. Conclusions

Our study demonstrates the therapeutic potential of deflamin to attenuate the development of pulmonary emphysema and peribronchial fibrosis in mice chronically exposed to ozone through interacting and regulating the catalytic activity of MMP-9. It also provides novel information on the use of deflamin to inhibit human MMP-9 in an in vitro assay and, through bioinformatic analysis, contributes to the understanding of its molecular interactions. However, further studies are required to determine whether deflamin also exerts an anti-inflammatory or antioxidant effect in a COPD model, as well as to evaluate pharmacokinetic and pharmacodynamic parameters.

## Figures and Tables

**Figure 1 ijms-25-05063-f001:**
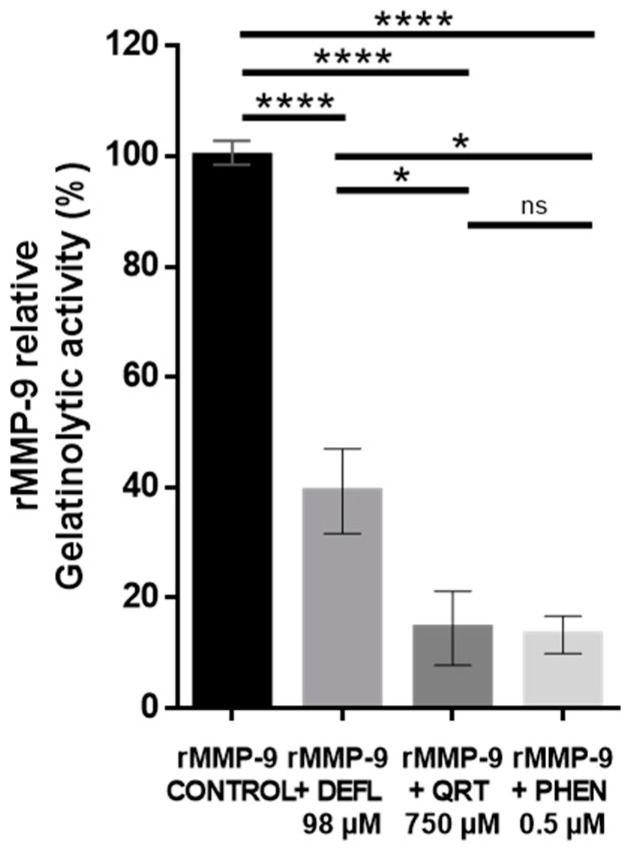
Inhibitory effect of deflamin on MMP-9 activity evaluated by fluorescent gelatin in vitro assay. *L. albus* deflamin was added to evaluate its inhibitory effect on the gelatinolytic activity of recombinant human MMP-9 (rMMP-9). Bars indicate the percentage of enzymatic activity relative to the control (rMMP-9 without inhibitor) and represent the averages of at least three experiments performed in triplicate. Results are expressed as means ± SEM. **** *p* < 0.0001; * *p* < 0.05; ns, not significant; DEFL, 98 μM deflamin; QRT, 750 μM quercetin; PHEN, 0.5 μM 1,10-phenanthroline.

**Figure 2 ijms-25-05063-f002:**
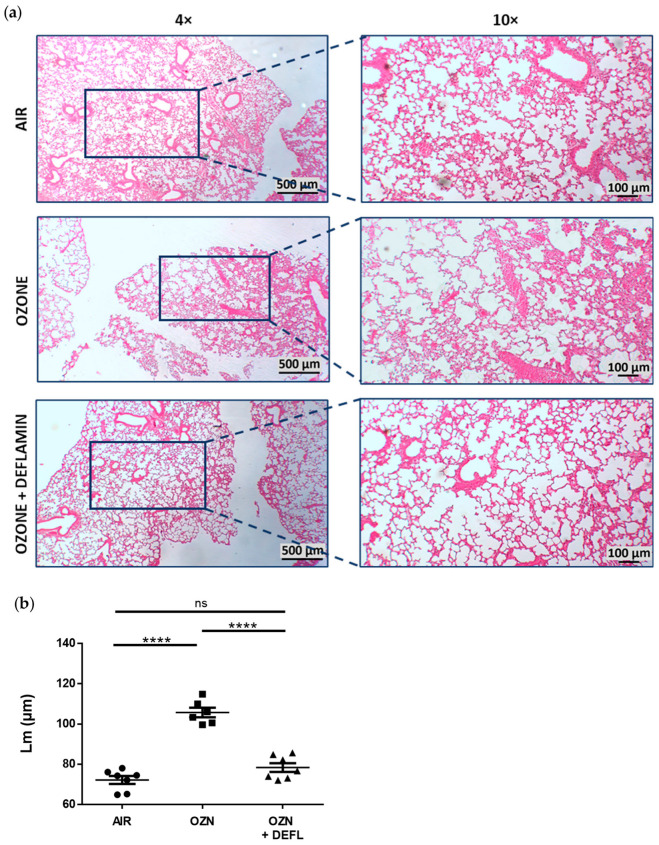
Comparison of histological lung changes in the experimental groups. (**a**) Representative photomicrographs from lung tissue stained with hematoxylin and eosin (H&E) at magnifications 4× and 10× that allowed us to assess the presence and severity of emphysema and vascular alterations among the groups. (**b**) Mean linear intercept (Lm) was quantified from the captured images and the results are expressed as means ± SEM. **** *p* < 0.0001; ns, not significant. OZN, ozone group; OZN + DEFL, ozone group treated with deflamin (20 mg/kg BW).

**Figure 3 ijms-25-05063-f003:**
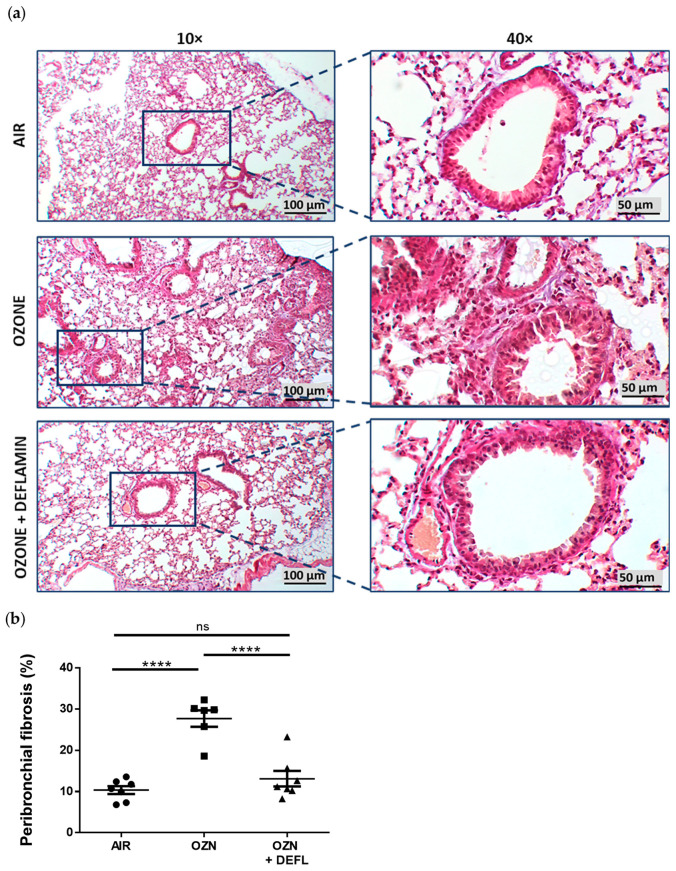
Peribronchial collagen deposition in lung tissues from the three experimental groups. (**a**) Representative photomicrographs from lung tissue stained with Masson’s trichrome staining at magnifications 10× and 40× to allow identification the presence and distribution of lung tissue fibrosis among the groups, especially in the peribronchial area. Collagen deposition is identified in blue–purple. (**b**) The percentage of peribronchial fibrosis was quantified from the captured images, and the results were expressed as means ± SEM. **** *p* < 0.0001; ns, not significant; OZN, ozone group; OZN + DEFL, ozone group treated with deflamin (20 mg/kg BW).

**Figure 4 ijms-25-05063-f004:**
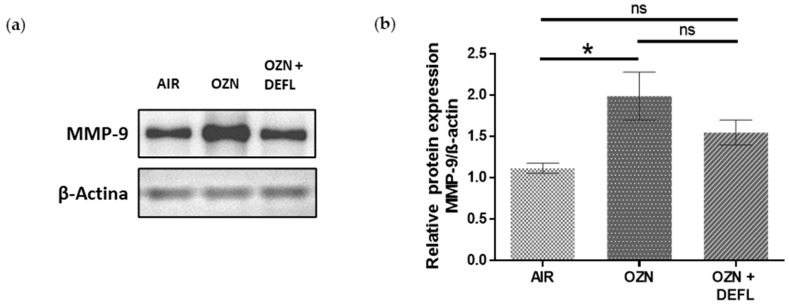
Matrix metalloproteinase (MMP)-9 protein expression in mouse lung tissue homogenates analyzed by Western blotting. (**a**) The protein bands corresponding to MMP-9 and β-actin are shown. (**b**) The quantitative analysis indicates a downward trend in the protein expression of MMP-9 in the group exposed to ozone and treated with deflamin compared to the ozone group. Data are expressed as means ± SEM. * *p* < 0.05; ns, not significant; OZN, ozone group; OZN + DEFL, ozone group treated with deflamin (20 mg/kg BW).

**Figure 5 ijms-25-05063-f005:**
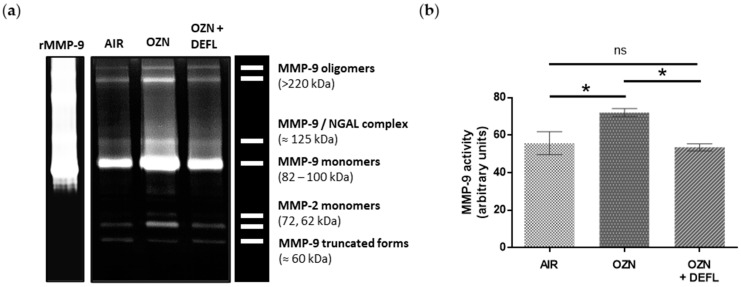
Matrix metalloproteinase (MMP)-9 activity of mouse lung tissue homogenates analyzed by gelatin zymography. (**a**) Representative gelatin zymography of lung tissue homogenates in the three experimental groups. (**b**) MMP-9 activity comparison (density arbitrary units) shows that deflamin statistically reduced MMP-9 activity in ozone-challenged mice. Data are presented as means ± SEM. * *p* < 0.05; ns, not significant; OZN, ozone group; OZN + DEFL, ozone group treated with deflamin (20 mg/kg BW); rMMP-9, human recombinant MMP-9.

**Figure 6 ijms-25-05063-f006:**
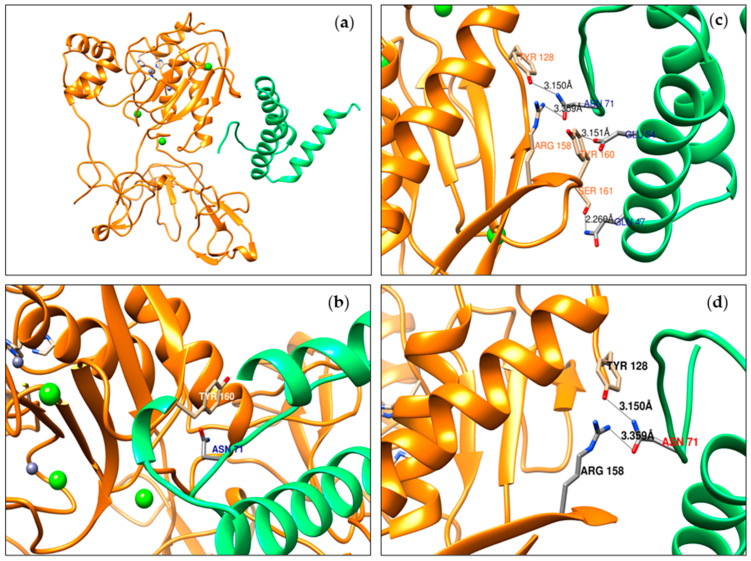
Three-dimensional molecular interaction between MMP-9 and the δ-conglutin large subunit. (**a**) Overview of the predicted interaction between MMP-9 (orange) and the δ-conglutin large subunit (green). (**b**) Close-up of the main interaction (no hydrogen bond) between TYR160 of MMP-9 and ASN71 of the δ-conglutin large subunit. (**c**) Close-up of the four main hydrogen bonds between MMP-9 and the δ-conglutin large subunit. (**d**) Close-up of two of the main hydrogen bonds in which ASN71 of the δ-conglutin large subunit was involved. Proteins are represented in their secondary structures. Calcium ions are in green and zinc ions in gray. Both the MMP-9 active site and the residues that are interacting, between proteins, are represented in sticks. Hydrogen bond distances are presented in Å and as black lines.

**Table 1 ijms-25-05063-t001:** MMP-9 and δ-conglutin large subunit hydrogen bonds (H-bonds).

MMP-9	δ-Conglutin Large Subunit	Distance (Å)
TYR 128 (C)	**ASN 71**	3.150
ARG 158 (C)	**ASN 71**	3.359
**TYR 160 (C)**	GLU 54	3.151
**SER 161 (C)**	**GLN 47**	**2.260**
ARG 162 (C)	GLU 46	2.772
ASP 163 (C)	ARG 43	2.863
ASP 280 (FN)	ARG 27	2.677
ASP 284 (FN)	ARG 23	2.938
TYR 311 (FN)	GLU 75	2.478

Highlighted in bold are TYR160 of MMP-9 and ASN71 of the δ-conglutin large subunit, which participated in 10 and 6 additional interactions, respectively. As well as the H-bond between SER161 and GLN47, which has the shortest distance between them. Inside the parentheses is specified whether the MMP-9 residues correspond to the catalytic domain (C) or to the fibronectin domain (FN). H-bond: hydrogen bond.

**Table 2 ijms-25-05063-t002:** MMP-9–δ-conglutin large subunit main electrostatic interactions.

MMP-9	δ-Conglutin Large Subunit	Distance (Å)
ARG 134 (5)	ASN 71 (7)	4.832
ARG 134 (5)	ASP 73 (4)	4.086
ARG 134 (5)	GLU 75 (4)	2.626
TYR 160 (11)	GLN 47 (5)	4.842
TYR 160 (11)	ASN 69 (5)	3.149
**TYR 160 (11)**	**ASN 71 (7)**	**2.797**

Of a total of 51 interactions between MMP-9 and the δ-conglutin large subunit, this list was obtained by identifying the residues of each protein that formed the highest number of different interactions. TYR160 and ASN71 are highlighted in bold, and were formed by the two residues with the highest number of different interactions for each protein. The number of interactions from each residue is indicated in parentheses.

## Data Availability

The datasets used or analyzed during the current study are available from the corresponding author on reasonable request.

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
