# Peer review of "Deflamin Attenuated Lung Tissue Damage in an Ozone-Induced COPD Murine Model by Regulating MMP-9 Catalytic Activity"

_ijms, 2024, doi:10.3390/ijms25105063_

Round 1

Reviewer 1 Report

Comments and Suggestions for Authors

This manuscript presents the effect of deflamin as an MMP-9 inhibitor. The authors showed the inhibitory inhibitory activity of deflamin against MMP-9 in some parts. This manuscript needs major revision prior to publication. Detailed comments are provided below.

Comments:

1.        Figure 1.

a.        The original gel data of MMP-9 with quercetin and phenanthroline.

b.        Why the control does not have error bars? Why the samples have only positive error bars?

c.        What are the gel lanes with “X”?

2.        Please keep the same format to indicate MMP-9.

3.        Why the authors have MMP-2 lane in Figure 5? It is not necessary for this study.

4.        For simulations,

a.        Why the authors use only delta-conglutin large subunit? What about other domains of deflamin?

b.        Can deflamin chelate out Zn(II) from the active site of MMP-9?

c.        What is the PDB number of MMP-9 the authors used for simulations?

d.        The authors tried to analyze the hydrogen bond formation between MMP-9 and delta-conglutin, however, what about other interactions? Were there hydrophobic, pi-pi interaction, or other interactions?

e.        How significantly the structure of MMP-9 has been modified by interacting with delta-conglutin? Can it affect the enzymatic activity?

5.        Figure 5

a.        The air and ozone treated samples have similar MMP-9 activity with consideration of error bars. It needs more discussion. The error bars indicate that the difference is not significant.

b.        Please provide negative error bars as well.

6.        The authors added 2.5 ppm of ozone, however, is that amount enough for causing lung damages?

Reviewer 2 Report

Comments and Suggestions for Authors

The paper “Deflamin attenuated lung tissue damage in an ozone-induced COPD murine model by regulating MMP-9 catalytic activity “ by Baltazar-García  et al., analyzed the effect of deflamin in COPD murine model.

The paper is interesting as it investigates how a polypeptide present in lupine seeds can mitigate this significant pathology.

The authors need to fix and elaborate on some points.

1. In vitro model lanes 93-108—The doses used of deflamin, quercitin, and 1,10-phenantroline are indicated in the materials and methods but should also be reported in the results to make reading the paper more fluid.

2. The doses of deflamin should also be reported in all figure captions.

3. The authors should explain the dose of or deflamin used; have any MM-9 dose-response curves been made?

4. The treatment of mice with deflamin is indicated in the methods. However, I recommend reporting them at the beginning of paragraph 2.2 because the reading becomes complicated if you must read the methods every time.

5. I would ask the authors to comment on the treatment because they take it for granted that deflamini is absorbed tout court at the intestinal level. Is it modified by digestive processes (enzymes, stomach acid pH, etc.)? Do the microbiota and liver enzymes modify it?

6. The molecular docking experiments between deflamin fragments (which one?) imply that deflamin does not undergo any modification before arriving in the blood, but I repeat, the microbiota and phase I and II liver enzymes do not modify it. How is it transported in the blood? Please, I ask the authors to comment on these points

Comments on the Quality of English Language

 Minor editing of English language required

Round 2

Reviewer 1 Report

Comments and Suggestions for Authors

All my concerns were cleared. Not this manuscript is suitable for publication.